# A Preliminary Technology Readiness Assessment of Morphing Technology Applied to Case Studies

**DOI:** 10.3390/biomimetics8010024

**Published:** 2023-01-06

**Authors:** Marco Fabio Miceli, Salvatore Ameduri, Ignazio Dimino, Rosario Pecora, Antonio Concilio

**Affiliations:** 1ALI—Aerospace Laboratory for Innovative components, 80146 Napoli, Italy; 2Department of Adaptive Structures, The Italian Aerospace Research Centre (CIRA), 81043 Capua, Italy; 3Department of Industrial Engineering, University of Naples “Federico II”, 80125 Napoli, Italy

**Keywords:** SMA, flap, winglet, morphing, finger-like, TRLs

## Abstract

In an innovative system, it is essential to keep under control the crucial development phases, which should consider several aspects involving, for instance, the modeling or the assessment of suitable analytical representations. Aiming to pursue a final demonstration to verify the actual capability of an engineering idea, however, some fundamental elements may have been partially considered. Many projects state the initial and final technology readiness level based on the famous scale introduced by the US National and Aeronautics Space Administration (NASA) many years ago and now widespread in many fields of technology innovation. Its nine-step definition provides a high-level indication of the maturity of the observed innovative system. Trivially, the resolution of that macroscopic meter is not made for catching advancement details, but it rather provides comprehensive information on the examined technology. It is, therefore, necessary to refer to more sophisticated analysis tools that can show a more accurate picture of the development stage and helps designers to highlight points that deserve further attention and deeper analysis. The risk is to perform a very good demonstration test that can miss generality and remain confined only to that specific experimental campaign. Moving on to these assumptions, the authors expose three realizations of theirs concerning aeronautic morphing systems, to the analysis of a well-assessed Technology Readiness Level instrument. The aim is to define the aspects to be further assessed, the aspect to be considered fully mature, and even aspects that could miss some elementary point to attain full maturation. Such studies are not so frequent in the literature, and the authors believe to give a valuable, yet preliminary, contribution to the engineering of breakthrough systems. Without losing generality, the paper refers to the 2.2 version of a tool set up by the US Air Force Research Laboratory (AFRL), and NASA, with the aim to standardize the evaluation process of the mentioned nine-step TRL.

## 1. Introduction

Within the scientifical and technological community, Morphing System architectures are very well-known and largely investigated. Realizing an aircraft system capable of behaving like a common bird wing has accompanied throughout the history of the evolution of airplanes. The vision of Clement Ader, [1], the first, almost legendary patent of Alexander Holle, [2], and the futuristic imagination of Mr. Parker, [3], were the first milestone and inspiration for all the researchers and engineers that looked at such an opportunity, almost naturally when dealing with the “heavier-than-the-air” navigation system.

Within the 1970s and 1980s, the futuristic “Adaptive Wing” project, was still the most impressive program on the adaptive wing maturation, which explored the possibility and the potential of installing shape-variation components on a military aircraft, the F111 fighter, [4,5]. Despite the success of the mission, and the demonstration of the potentiality of the technology, the system was not applied to the final aircraft. The technology was not mature enough to guarantee the lightness, easiness, and cheapness, as well as the correlated safety levels, necessary for a system to fly on an operational aircraft. A new impulse came in the late 1990s, almost close to the new millennium, with the DARPA-sponsored fascinating research on the applicability of “Smart Wing” technologies, [6]. The authors believe that this new exploration was mainly linked to the appearance on a wider technology scenario of the shape memory alloys (SMA), which peculiar properties were discovered by William Buhler in 1967, [7]. The fact is that Jay Kudva, the principal investigator of that activity, strongly relied on the use of SMA, in his first trials, [8]. However, he concluded that this kind of technology was not mature enough to be exploited on an actual aircraft (for many reasons whose discussion would go definitely beyond the scope of this paper), and it was more effective to implement contemporary compact actuators technology for attaining the prescribed objectives, [9].

That may be considered the real turning-point of the research on the adaptive wing thematic, when the number of publications, theoretical developments, and large-scale experiments literally exploded [10]. Perhaps, the most relevant application examples were: ACTE, Adaptive Compliant Trailing Edge, a result of the cooperation among NASA, and AFRL which led to a flight test on a Gulfstream Citation of an adaptive flap, implementing a technology developed by Flexsys, [11]; the Adaptive Trailing Edge program, a cooperation between Boeing and FAA, that led to flight test smart actuators based on SMA to move a kinematic chain of a small flap, [12]; VCCTEF, Variable Camber Continuous Trailing Edge Flap, a different another project of Boeing aimed at developing a new class of adaptive systems still placed on the backward termination of the wing, [13]. In Europe instead, the SARISTU project was certainly a major milestone on the technology development, with three different morphing systems mounted on a full-scale wing section and tested in wind tunnel at Tsagi, [14]. Another important ongoing European project, AG2, mean at testing in flight on regional aircraft, an adaptive winglet, and an adaptive wingtip, in the course of the next year (2023), [15].

However, although huge efforts have been made, the scientific and technological community has started wondering why morphing is not mature for practical implementation yet, at least on larger aircraft. In the present work, the authors intend to analyze three morphing wing assemblies, chosen between their most representative realizations developed during past and ongoing research projects, by assessing their technological level and identifying the related technology leaks, [16,17,18]. The output of such a study could be fully considered representative of the status of technology and provide important hints for their further development. How this investigation can effectively evaluate the development of a certain technology, and how it can address or validate the future steps of the evolution of that technology are the objectives of this paper. This manuscript provides an original analysis of advanced morphing systems to identify potential technology needs to make the addressed systems suitable for commercial production and implementation on mass aircraft.

To address this specific study, it was necessary to establish a common, and possibly, widespread method to have a measure of the achieved TRL (Technology Readiness Level), [19,20,21,22,23]. The methodology proposed by the AFRL was then selected, i.e., the Technology Readiness Calculator, as it seemed comprehensive enough to give a full overview of the attainments and the possible leaks of a general system. Furthermore, the links with NASA gave the confidence that the approach can be potentially shared and discussed among a wide user community, [24,25]. Although this tool is widely considered as a consolidated method to provide a snapshot of technology maturity at a certain time, it should be also considered that the related outcomes may not necessarily fit the macroscopic expectations. In fact, significant variability in the TRL assessment of the single subsystems may occur, for instance depending on the maturity of the application environment at hand, which is in turn due to a subjective understanding of the complex pathway enabling technology maturation.

To carry out this work, it was necessary to have full access to the details of each technology under investigation, so that the TRL measure could have been reliable to the maximum extent. The selected systems are different for complexity, logic, layouts, and aims. The first one is a multifunctional adaptive flap, capable of different ways of working within the same architecture and the same physical realization, all congruent with the morphing and the adaptive structure definition, [26], to meet different flight phases necessities (cruise, take-off, and landing, maneuver). The second system is an adaptive winglet, [27,28], and is made of the main body with external independent multi-tab systems capable of working both synchronously and asynchronously, for performance adaption among different flight phases (take-off, cruise), and for load alleviation purposes. Finally, the third one implements a full-embedded SMA torque tube for twist adaptation of a full-scale rotorcraft blade for matching both hovering and cruise needs, [29]. The compliant architecture and the use itself of smart materials were almost a constraint, to match the limited room available and weight tolerance allowed.

The paper is structured as follows: Chapter 2 is devoted to a general introduction of the investigated morphing systems, giving basic details, and references for further insights. Chapter 3 introduces and details the selected TRL measurement tool. Chapter 4 focuses on the application of that tool on the referred systems, providing a critical discussion on how the results were obtained. Chapter 5, on the basis of the attained outcomes, provides a detailed discussion on the specific morphing systems, trying to exploit those particular results to a more general prospective, and pointing out the technological needs (whenever necessary) for their massive implementation on large aircraft systems.

## 2. A Short Description of the Analyzed Morphing Systems

In the following section a brief description of each referred morphing systems, and their related technologies, is reported. The related state-of-the-art and the work done will be presented to introduce the framework of the TRA (Technology Readiness Assessment) carried out in the present work.

### 2.1. Adaptive Flap Description

The adaptive flap was developed within the framework of the CleanSky and CleanSky2 programs, [30]. The device is characterized by a multi-box arrangement with shape-morphing capabilities enabled by robotic (finger-like) ribs driven by electro-mechanical actuators. Three different morphing modes are enabled depending on flap settings/aircraft attitude:Morphing mode 1. Overall camber morphing of the flap cross-section, to enhance aircraft high-lift performance during take-off and landing (flap deployed);Morphing mode 2, Tab-deflection. Upwards and downwards deflection of the flap tip in the range [−10°: +10°] during cruise, to enable wing load control at high speed (flap stowed). The flap tip chordwise extension is equal to the 10% of the flap chord (skinned region in Figure 1) and represents the portion of the flap still exposed to the aerodynamic flow when the flap is stowed in the wing.Morphing mode 3, Tab-twist. Differential deflection of three consecutive segments of the flap tip (tabs 1, 2, 3 in Figure 1) always in the range [−10°: +10°] per segment, to further enhance aerodynamic load redistribution along the wing at high speeds (flap stowed).

Each flap rib consists of four consecutive blocks (B0, B1, B2, and B3) hinged along the camber line at the positions marked by points A, B, and C in Figure 2. Block B0 and B2 are further connected through a link, denoted with L in Figure 2, which makes B0, B1, and B2 to move as a single degree-of-freedom mechanism. The last block, B3, is connected to block B2 only. Two rotary electromechanical actuators (A1, A2, in Figure 2) are adopted to morph the rib; the first actuator (A1) drives the motion of the first three blocks, while the second actuator (A2) controls the rotation of the last block around the hinge C. The torque of each actuator is duly amplified by optimized mechanisms (M1 and M2, in Figure 2) fully integrated in the robotic rib.

Owing to a through-shaft solution, each couple of actuators drives three (/two) consecutive ribs (/bays) of the flap (Figure 1). During morphing mode 2 (/3), the amplification mechanism M1 is locked through permanent magnetic brakes, and actuator A1 is powered off; the actuator A2 is then activated, and the torque amplification mechanism M2 induces the upwards/downwards deflection of the rib tip (i.e., the block B3); the actuator shaft rotation and the induced rib tip deflection are constant along the span for mode 2, and step-wise variable along the three flap tabs for morphing mode 3. During morphing mode 1, both the actuators are simultaneously activated; the torque of the actuator A1 is used to move the amplification mechanism M1, which in turn makes the block B2 rotate with respect to the hinge B; since B0, B1, and B2, together with the link L, make a single degree-of-freedom mechanism, the rotation of B2 around B forces the rotation of B1 around A, according to specific gear ratios. The actuator A2 is synchronized to actuator A1 to properly control the deflection of the rib tip according to the positions assumed by blocks B0, B1, and B3, and the morphed shape of the cross-section to be implemented.

The skin of the flap is made of segmented panels, covering each chordwise portion of the device, Figure 3: one panel for the leading edge, two panels (upper/lower) to cover the first and second chordwise block of the flap, two panels (upper and lower) to cover each tab. Skin panels slide along each other in chordwise direction to accommodate the change of shape, rubber seals are used at the boundaries of each panel to prevent flow leakages. The entire flap structure is made in light AL-alloy with the exception of some elements of the rib internal mechanisms made in steel.

Starting from the reference aero-loads related to the most severe working conditions expected in service, [30], the morphing flap device was designed according to an articulated engineering process compliant with CS-25 requirements and industrial standards [30,31]. To properly monitor the process, three revision gates were placed at the end of the conceptual, preliminary, and advanced design phases (critical concept review, preliminary design review, and critical design review). The goodness of the adopted structural solutions as well as of the actuation and control strategy were successfully proved by means of full-scale static, functionality, and dynamic tests, Figure 4, [30]. Relying upon the outcomes of the dynamic tests, a thorough assessment of the flap impacts on aircraft aeroelastic stability was also addressed, showing the absence of any flap-induced flutters also in case of flap malfunctions (jamming) and/or free plays, [31]. In parallel, the aerodynamic improvements brought by the device at aircraft level were successfully demonstrated by means of large-scale wind tunnel test, fairly reproducing flap’s low speed (take-off and landing) and high-speed (cruise) operative conditions.

### 2.2. Adaptive Winglet Description

Within the framework of the Clean Sky 2 REG IADP, the design of a fault-tolerant adaptive winglet concept was addressed to enhance the wing’s aerodynamic performance in off-design conditions and reduce maneuver loads of a turboprop regional aircraft. The integrated design of the adaptive winglet is detailed in [32,33,34]. The adaptive winglet incorporates two “finger-like” mechanisms, shown in Figure 5, allowing shape adaptation of two movable surfaces (upper and lower tabs), each controlled by an independent electromechanical chain, and embedded into its main body [27,35,36]. The concept was based on the following, general assumptions:Morphing winglet system chord = 40% of the mean winglet chord.Deflection range = [−15°: +5°] (negative values indicates RBM reductions).

Both the spars and ribs of the winglet structural torsion box are made of carbon fiber laminate; instead, movables are made of aluminum alloy due to the tight tolerances therein requested. Actuation is accomplished by two electromechanical actuators of adequate size, weight, and power, that are housed inside the CFRP structure. Each mechanical system has two separate hinge lines enabling the camber morphing of the two devices through individual mechanisms, fitted within the confined space of the winglet loft-line. Both actuators dimensions and morphing mechanisms were optimized to achieve the best compromise between motion amplification and load capacity. Figure 6 shows the finger-like mechanism developed for the upper morphing tab. Each tab is a single-degree-of-freedom kinematic system where the rigid blocks rotate according to a specific gear ratio, depending on hinge and linking rods position.

The aerodynamic design of the adaptive winglet was evaluated by CFD simulations. A significant increase in LoD, estimated on the order of 2.5%, was obtained at high CL (climb conditions) with respect to the optimal passive winglet counterpart designed for the same aircraft. On the other hand, a limited aerodynamic benefit was observed for small positive deflections at cruise conditions, being this condition one of the design flights point of the wing. Figure 7 shows an example of morphing winglet tabs deflection of 10 deg.

Following the technology maturation roadmap of the device, the full-scale demonstrator of the adaptive winglet was manufactured, and ground tested in July 2021, Figure 8. The experiments consisted of three different phases. First, a static test was performed on the winglet structure (without the kinematic chain and the morphing parts) bolted to a static test rig in order to assess the loading system and the test rig deformation at 30% of the limit load. A whiffle tree was used for load distribution from load cell to the distribution pads connected to the skin of the winglet in order to simulate the external applied loads. Both the load levels to be introduced to the winglet and loading positions were numerically calculated and alternatively applied to the fixed structure and the two moveable parts. The magnitude of the forces to be introduced to the winglet were defined such as to represent the stress state expected during flight. Strain gauges were mounted on both the structural parts of the winglet and the kinematic chains of the moveable parts. Their strain values were recorded by a dedicated acquisition system. In compliance with the applicable sections of airworthiness requirements, such a test campaign proved the ability of the winglet to withstand limit loads without any local plasticization or elastic instability. Moreover, the morphing tabs were tested by discrete loads and static displacements and strain values were measured on the corresponding skin and the kinematic chains, respectively. For that purpose, the morphing tabs were blocked in requested position by a rigid bar simulating the actuator’s stiffness. Two dedicated whiffle trees connected to the load pads were used for pressure loading the moveable surfaces with the appropriate load. Ultimate loads were not tested due to cost and time limitations of the project. Finally, functional tests were carried out to demonstrate the actual capability of the morphing tabs and related actuation chains to replicate the target aerodynamic shapes with and without the effect of aerodynamic loads. Such tests were performed to characterize the behavior of the movable parts, and to verify the internal hinges and mechanical links connecting the morphing parts. Both shape and strain measurements were taken in unloaded conditions in order to characterize the nominal aero-shapes produced by the winglet, to be compared with the numerical predictions. After that, functional tests under static loads were executed. Two linear electromechanical actuators with limited load-bearing performance were integrated to control the morphing mechanisms of the two devices. In such a loaded condition, the two morphing devices were then activated and both the achieved shapes and strain responses were measured. They were simultaneously recorded to assess the actual morphing capabilities of the physical demonstrator and characterize any potential jamming or interference phenomena that could prevent the designed moveable from assuming the desired shapes. The acquired morphing shapes were then compared with those acquired in unloaded condition in order to assess any deviation from the nominal mechanical behavior due to the external loads. Such differences can cause aerodynamic performance degradation and may impact the aircraft controllability that has to be addressed during system design. Such an assessment was done for increasing values of loads from 0% to 100% of the limit load (LL) and including 33% and 66% LL.

### 2.3. Shape Adaptive Blade Description

The SMA blade twist concept was developed within the Project “Shape Adaptive Blades for Rotorcraft Efficiency” (SABRE) [29], under the H2020 framework. CIRA developed a morphing system that was able to change the blade twist in compliance with the targets of the Project: the elaboration of adaptive technologies affecting the main rotor features to improve the helicopter performance. The core of the architecture consists of a compact and solid shape memory alloy (SMA) rod, used as torque/twist actuator, [39,40]. Embedded along the span, and connecting two cross-sections, such a driving device is activated by a thermal field, generated by a heating coil wrapped around the SMA and enforcing the phase transformation from martensite to austenite, with the macroscopic effect of recovering a pre-imposed deformation. In Figure 9, the main parts of the architecture are shown.

A dedicated integration process was implemented to realize the morphing system, divided in the following phases (the final product is shown in Figure 10):*Training of the SMA rod*: (a) several heat-cooling cycles were carried out to stabilize the mechanical alloy response, and obtain the required thermal stress relief; (b) torsional load-unload cycles were then addressed to stabilize the actuation performance.*Pre-twist of the SMA actuator*: a side of the SMA rod was clamped to the solid structure, while the other one was connected to the gear torque element (part D in the figure).*Full connection to the solid structure:* (a) at the end of the pre-twist phase, dedicated holes on the gear element were aligned to corresponding ones on the structural element; (b) pins were inserted to make the components integral to each other; (c) pre-twist moment was removed, and let the system to achieve natural elastic equilibrium.*Rib and balancing masses installation:* the rib components and the balancing masses are linked to the solid structural component.*Sensor system installation:* inclinometers and sensor supporting plates were mounted on the ribs.*Skin installation:* the exterior skin was inserted and screwed to the main structure.*Bays connection:* according to the modular nature of the envisaged blade system, each bay was connected to the other ones, through lateral plugs.*Cabling installation:* cabling runs through the ribs, up to the root.

A three-bay blade segment is shown in Figure 11. In line with the scope of SABRE and with the target of assessing the TRL of the concept, a specific experimental path was followed to demonstrate the actual capability of the manufactured system to produce the prescribed twist in representative environments. In detail, three different test campaigns were carried out. Laboratory tests were conducted to characterize the system early by simulating two reference configurations: fixed and rotary wing, respectively. In the first one, the blade demonstrator was installed in a clamped-pinned layout, so to allow the twist, while supporting both the ends of the system. In the second one, the blade demonstrator was camped at one edge, while an axial load was applied on the other side, replicating centrifugal loads occurring during rotation. In both the cases, hanged weights simulated the effect of inertia and aerodynamic forces. Twist produced by the SMA actuators was measured, with and without external loads. One or more bays were activated. To achieve deformed geometry reconstruction, morphed shape was related to the deformation field, retrieved by strain gages and fiber optics. The tests proved the capability of the system to produce the desired twist while withstanding the most severe load conditions with respect to the envisaged tests in wind tunnel (WT) and whirl tower (WHT). In fact, the single bay achieved twist angles of 0.68 and 0.55 deg for fixed and rotary configurations respectively that, scaled over the entire blade radius of 5 m, correspond to 15.3 and 12.4 deg, well higher than the requirement of a twist of 8 deg. WT tests aimed at measuring the aerodynamic performance in the unmorphed and morphed configurations, respectively, by activating progressively all the bays along the span. The blade was mounted in clamped-pinned configuration. Tests were run within the established angle of attack operation range [-11; +11] and at velocities of 95 and 120 rpm. Those tests confirmed the system capability in enhancing the aerodynamic performance. WHT tests targeted to evaluate the influence of the centrifugal forces on the production of the desired twist. In that case, the blade segment was simply clamped on one side and tested at different angles of attack, at the maximum speed compatible with the facility safety requirements. Even when turning at high speed, the morphing system proved to be capable of producing the required twist, therefore overcoming the potential obstacles linked to an increase of friction forces, structural stiffness, and the insurgence of certain architectural distortions. As shown in the bar plot of Figure 12 representing the actuated twist at the tip, scaled to a 5 m spanned blade, the morphing system produces the highest twist of 12.1 deg for the maximum AoA and velocity and a twist of 7.1 deg for the minimum velocity and at 0 AoA, slightly lower than the requirement (8 deg). In addition to that, the mass distribution together with the specific aerodynamic loads generated along the span, favor the twist transmission with the speed and the incidence.

## 3. The NASA/AFRL Technology Readiness Level (TRL) Tool

### 3.1. General Introduction

Standing to common definitions, as in [41], it may be relevant to distinguish between TRA (Technology Readiness Assessment) and TRL. The first intends to be a systematic process to define the level of knowledge required, including technical and risk aspects, for inserting a certain technology into a generic system, either novel or already assessed. TRL is a method to estimate the technology maturity of a certain technology, within a development project or program. Usually, it is preferably used for the so-called critical technology elements (CTE), focusing the attention of the developers on remarkable items, rather than applying it to everything. Implementation of TRL allows uniform discussion of maturity among different kinds of technology. In this paper, a TRA process for establishing the TRL of three different critical technology elements, all implementing aeronautical morphing concepts, has been conducted through the tool assessed by the AFRL and NASA, the TRL Calculator, Version 2.2 [19].

In the introduction to the Ver.I.2beta of the tool, James W. Bilbro from the AFRL gives a comprehensive and critical analysis of the needs and the aims of the product the team developed, the “TRL Calculator”. The following text is a short recap of his words.

Its realization was driven by the shared understanding that immature technology could have a tremendous impact on releasing products on time and within costs; at those times, a precise definition of maturity, and even of technology was not available. Technology development may be defined as “*the application of scientific knowledge to do something completely new or in a completely new way*”; as such it should be avoided in the development of any program/ project that has deliverables at fixed cost and schedule. Technology development may be framed in the so-called region of “unknown unknowns”, a domain outside of the common experience base. Many research program/ projects with those characteristics were often caught unaware, leading to budget overruns, schedule slips, and sometimes, even failures. The main cause may be a lack of up-front investments to setup and understand proper requirements and what is needed to meet them. If this consideration is valid for ex novo items, it holds even for heritage systems i.e., something that is already in operation: as a modification is applied, adjustments may be needed (and usually are), which are found outside the common experience. A basic rule is established and commonly applied, making the TRL of heritage systems drop to TRL5, typically, unless something different is demonstrated.

### 3.2. A Tool Description

The AFRL/NASA TRL Calculator is a relatively simple, systematic method for assessing the maturity of systems. In few words, it is a reasoned guidance, to assess the TRL of a generic technology product. It allows presenting the results together with the hypothesis that have led to them, favoring discussions, interactions, and the judgment share. The tool may be used at the very early stage of the project to provide a reference maturity assessment, as well as to measure progress and provide bases for a TRA report.

In the used form, the tool is presented as Transition Readiness Level, and is composed of three different parts, concerning the technology (the original TRL), the programmatic or strategic (named PRL, programmatic readiness level), and the manufacture assessment (manufacturing readiness level, MRL).

In its complexity, the architecture and the user interface are very simple. Each level presents a series of questions to be answered, concerning different aspects of the technology development, ranging from the publication of the results on peer-reviewed journals, to the issue of suitable specifications from the client (if any). Each question may be answered in a scale moving from 0 to 100%; there is a possibility of declaring “accomplished” in a certain level, without moving into further details. This is useful for sound or “heritage” attainments, but should be used with care, since this option does not guarantee the operator toward point questions from a generic reader, an auditor, or a reviewer. Questions concerning different TRL levels may be faced and accomplished, so that they are not filled sequentially but in parallel, and that the filling level may have the face of half a Gaussian or a sigmoid, instead of a step function.

In synthesis, the summary and the rationale of the different questions may be reported as follows:TRL: indicates the level of capability of the envisaged system to respond the issued requirements (engineering capability).PRL: indicates the clarity of the environment within which that technology is developed (actual applicability).MRL: indicates the possibility of realizing the individuated technology within industrial processes (i.e., no one-shot realizations: production feasibility).

Furthermore, each of the three levels may be separated for the hardware and software parts:HW: is referred to the capability of materially building the technological item.SW: is referred to the capability of controlling and guiding the item.

To have a quick impression of the tool architecture, Table 1 reports the number of issued questions, classified per type.

### 3.3. Tool Peculiarities

The authors of the tool recognize that it is necessary to discriminate among different engineering realizations and allow a certain subjectivity in the evaluation process. Nevertheless, such degrees of freedom are documented and tracked, so that each choice is duly registered and constitutes an integral part of the assessment. Since it could happen that some higher criteria may be matched before lower criteria do, it is required that the latter ones are satisfied before moving on. This shrewdness avoids neglecting essential steps in the development of the technology, so that solid background bases are always guaranteed. Therefore, it is not just a matter of making the system to work, but even to understand how it works, and to what extent.

## 4. Technology Readiness Assessment of the Presented Technologies

Establishing the maturity of a given technology is always a tricky process. It could be affected, either positively or negatively, by the subjective judgement and the experience of the evaluator. However, taking advantage of a well-defined and calibrated tool, the analysis is made more objective, thanks also to the specificity of the requirements needed to claim a given Technology Readiness Level. The present work aims at providing a Technology Readiness Assessment (TRA) for three different morphing systems, devoted to specific targets, using the AFRL Transition Readiness Level Calculator Rev. 2.2. A discussion on the achieved TRL (technology readiness level), PRL (programmatic readiness level), and MRL (manufacturing readiness level) is reported, together with a critical comment on possible future actions for the further improvement of their maturity. In that case, only hardware criteria have been considered. The same parameters were set, so as to ensure a certain comparability of the analysis outcomes.

### 4.1. TRA for Adaptive Flap

The wing flap camber morphing was developed within the Airgreen 2 project, and was therein tested during an extensive experimental campaign in a relevant environment, [31]. In the future, the prototype will be tested on a surrogate platform of the target system. Therefore, the current TRA may be preliminarily assessed to be 5. In Figure 13, the effort that has been spent so far is shown in terms of number of fulfilled requirements for each level of the used tool. As anticipated in the paragraphs above, such analysis relies on the engineering experience and sensibility of the designer; nevertheless, the process is documented and ready for further reviews. This approach is applied for all the other evaluations, and is further presented in the paper.

Going into detail, minor actions are needed to set the PRL from 3 up to 5, which may be considered the expected value. The main pending action is the establishment of an integrated product team (IPT). In any case, the technology may be said to has reached a good level of maturity, from a programmatic point of view. In fact, the technology owner successfully implemented the defined specifications, as verified in dedicated PDR (preliminary design review), and CDR (critical design review) events. Furthermore, the development of the actuation system faced the reliability consolidation, and the inherent reduction of the parts number [26]. Therefore, the decrease in realization, integration, and maintenance costs has been considered, even though it was not yet formally quantified and drafted in documentation yet, with respect to a conventional technology. The completion of the aforementioned steps would bring the overall PRL to 5 (at the 67% of completeness). Other programmatic paperwork and evaluation should also be addressed: for instance, a formal CAIV (cost as an independent variable) targets setting, a formal risk management program, and a dedicated values analysis, that include life-cycle costs analysis. Those documents are also vitally propaedeutic for the effort that should be spent to achieve the TRL 6 in the further advancement. The steps needed to fully achieve the 5th level of the PRL addressed by the Industrial partner of the technology owner. A dedicated business case analysis in which the so-called Life-cycle-costs of the developed technology are compared to those of a conventional one, should be addressed. The analysis should also include a Roadmap with foreseen industrial developing steps, where at least an advanced technology demonstration and the availability of the technology on the market should be identified.

MRL and TRL may be quantified to 4, with the necessity of implementing few further actions to complete the level 5. Test campaign outlined that the designed assembly is fully representative of the actual system both in function and form. Therefore, it is possible to state that most of the effort to achieve higher level of maturity was already spent so far. A synthetic graphical vision of the technology readiness assessment status is reported in Figure 14, to allow the reader an immediate perception of the current status. Therein, the amount of achievement is normalized with respect to the number of actions requested to claim a generic overall readiness level for each of the section proposed by the tool (i.e., TRL, MRL and PRL), according to its original release.

### 4.2. TRA for Adaptive Winglet

The basic concept behind the architecture of the winglet morphing system is based on the same principles of the multifunctional flap ones, implementing a finger-like mechanism, [28]. As the former, it has been developed within the Airgreen 2 project. The technological capability has been widely addressed and extensively documented, [27,32,33,34,35,36]. A full-scale prototype of the functional winglet was tested in a relevant lab environment, under the design limit loads. Therefore, the technology may be preliminarily assessed at a TRA equal to 6. Figure 15 shows graphically the effort that has been spent so far for each level of the used tool.

A more detailed analysis, referring to the TRA tool, allows discovering some interesting aspect of this morphing technology. The implemented actuation device is mature, based on commercial products. The winglet has been endorsed by a manufacturer, and potential customer of the device. They participated in the device development, within an established IPT. That team defined suitable design, scaling, and risk mitigation strategies. An exhaustive finite element model (FEM) analysis has been carried out, aiming at improving the definition of the adaptive system, characterizing technological capacity, and contributing to the risk assessment [32]. A safety analysis was also performed to explore the failure modes of the device [28]. The PRL may be then assessed at 5, with very few steps further needed to get the level 6.

Concerning the MRL, it can be stated that level 5 is next to be fully achieved, while a remarkable amount of level 6 issues has been already faced and assessed. Among the relevant still pending issues, there are: a study for defining sigma levels and assessing the CAIV, or a complete review of the production processes, to be carried out in cooperation with the producibility, quality, and manufacturing offices.

To be complete, TRL needs some devoted, specific efforts, Figure 16. Concerning the level 4 of the used tool, a formal document on cross technology issues is pending, and both acceptance and sub-assembly test campaigns are ongoing at the time of this survey. Closing those actions would bring the TRL very close to level 5/6.

### 4.3. TRA for Shape Adaptive Blade

The activities related to this device were carried out within the SABRE project, and involved extensive test campaigns on three different system breadboards based on consolidated knowledge of SMA-based actuators [40,42,43] and [44,45,46]. In detail, a wide FEM analysis, [42], and mechanical characterization of shape memory alloys-based systems were conducted in suited configurations, [44]. The latest experiments aimed at evaluating the system performance in a relevant environment, installed on a surrogate of the hosting platform, and implementing due modifications for refining its design. Therefore, the attained TRA may be set to 6, preliminarily. This synthetic evaluation is graphically reported in Figure 17.

Going into a detailed analysis of the referred tool, the further accomplishments that are necessary to consolidate the attainment of that level emerge.

The current PRL should be considered at around 2 since some missing goals prevent to reach a higher level of maturity. The technology development has been endorsed by a large manufacturer, but the scaling strategy, together with an investment strategy sheet, is further assessed by an independent advisory board. Similarly, the design requirements address the implementation of the addressed technology on an actual, operating system. According to the available product documentation, a system engineering master plan (SEMP), and a test and evaluation master plan (TEMP) are completed at 40%. It is estimated that the issue of those documents, together with the issue of a risk management plan (RMP) alone, could boost the overall PRL up to 6.

The extensive test campaign carried out within the project, allows this morphing technology reaching a very high level of fidelity with respect to the final system. The Assembly, Integration and Test campaigns permit to complete the MRL grade 6 of the reference TRA tool, at 67%. Given the proven difficulty in machining NiTiNol alloys, [46], an analysis on the manufacturing process of the SMA rods should be addressed in the future. In perspective, this analysis is required for the achievement of MRL 7, where manufacturing process, tooling, design techniques represent more than the 50% of the overall effort to be spent.

The same level of completeness is reached by the TRL, with some open issues. For instance, the effect of the SMA/monobloc interface is not well established yet; even though this interface exhibited an excellent behavior, each adaptive bay performed slightly different from another, highlighting the need for further investigations.

By integrating the different considerations herein drafted, it may be stated that the device is safely assessed from a technology and a manufacturing point of view, while it needs some dedicated effort to consolidate its production standards. Such a situation is reported in Figure 18.

## 5. Impact of the Assessment on Future Technologies’ Development

### 5.1. Impact on the Multi-Functional Flap Development

The morphing flap technology has been assessed at TRL5, relying upon the outcomes of the true scale experimental campaign carried out on the ground, and fully validating the theoretical expectations regarding morphing performance and structural robustness. The structural and mechanical configuration of the system is characterized by an acceptable level of complexity, well justified by the number of smart functions enabled by the device. From the weight breakdown standpoint, the morphing flap mass slightly exceeds the typical values of the conventional flaps; however, if the attention is shifted to the overall high-lift system, intended as flap and related deployment mechanisms, then the adaptive technology gains several points of advantage on the traditional one. The morphing flap can be placed at a unique setting when out of the wing; the shape of the airfoil can then be changed to get the high lift required by different flight phases (take-off and landing, mainly), always in correspondence with the same flap setting. This paves the way for a dramatic simplification of the deployment system, with a consequent reduction of its weight. As a matter of fact, the extra weight of the adaptive flap is fully counterbalanced by the weight saving on the associated deployment system, thus leading to an overall weight of the high-lift system entirely in line with the standards of those already in service. To bridge the gap with already flying technologies, targeting TRL 6 and beyond, other research activities are, however, necessary, mainly along three different lines and with an impact on various sub-components of the device, Table 2:Morphing flap-aircraft interfaces;Serialization of the prototype;Flight safety.

So far, only the mechanical interfaces between the flap and the wing have been investigated in detail. Therefore, specific attention should be paid to integrating the actuation and sensing systems into the aircraft equipment at both hardware and software levels. Research efforts need to be spent along the path of the electrical interfaces, cables-routing, and communication protocols between the cockpit and the device. The geometrical layout of the flap subcomponents and the joints between different subcomponents are another aspect that deserves more detailed analysis; although the flap’s architecture has been designed while duly taking into account the impacts of the adopted solutions on assembly/disassembly time, a further design optimization loop must be carried out in perspective of series production, inspection, and continued maintenance. Clearly, well-defined installation and operative frameworks represent the fundamental assumptions to address these tasks adequately. The serialization of the flap prototype must be conceived in combination with the manufacturing and assembly process of the entire aircraft the flap will be installed into, being deeply influenced by its peculiar approaches, fabrication methods, and duration. The same considerations also apply to the design optimization aiming to minimize the impact of the novel components on aircraft inspection and maintenances approaches.

Finally, the road toward flap flight testing and entrance into service passes through the solution of important safety issues that have not been addressed yet. In principle, the entire structure has been designed in compliance with the airworthiness requirements considered relevant for the ground demonstration of the prototype. The compliance with further requirements of EASA CS-25 [47], must be shown to push the technology readiness level to the next levels. Since the flap is used also as load control item, appendix K of the regulation needs to be addressed, and the adequate redundancy of both structural and electrical items must be defined to increase the safety of the smart system in operative conditions. In line with this target, the fatigue life assessment of device represents an additional paramount activity to be performed especially considering the unconventional nature of the structural architecture and embedded electrical systems, potentially leading to novel failure modes and/or malfunctions. The adoption of an intelligent sensing system monitoring both the shape of the flap and the level of integrity of all its subsystems may represent a key solution in the prevention of fatigue-related problems, thus speeding up the process for a more conscious adoption of this morphing technology into civil air transportation.

### 5.2. Impact on the Adaptive Winglet Development

The adaptive winglet technology has been assessed at TRL 6. Due to the overall project’s goals and the tight requirements for the full-scale validation and demonstration of the technology, a relatively complex kinematic concept was developed for the two morphing winglet trailing edges, that met very strict tolerances but resulted in a very high part count. This in turn led to the need for advanced FE simulations supported by dedicated multi-body analyses aiming at facilitating the integration activities in a suitable assembly process. An industrially feasible concept was then matured based on a suitable system design capable to meet the stringent CS-25 requirements by ensuring at the same time a tight geometric control of the morphing parts in operative conditions. The main design aspects and relevant potential approaches identified to increase the proposed technology TRL are listed in Table 3.

The design and manufacturing of the full-scale winglet demonstrator that still lacks a reliable production process to be considered as MRL 6 were centered around the challenge of locating two electromechanical actuators and the related actuation kinematics inside the device’s loft-line, without affecting the optimal aerodynamic profiles validated by high-fidelity CFD simulations. This was complicated by the very limited space available at the winglet root to accommodate commercial actuators and the relatively high hinge moments resulting from the design loads acting on the moveable parts. Overall, the actuation kinematics can be considered mature enough to be deployed in actual applications and, most importantly, the comprehensive failure and hazard analysis performed during the project, with input from industry, showed that no critical safety concerns are associated with the actual implementation of the technology.

As a result of a de-risking strategy, the use of a segmented skin concept, consisting of separate skin blocks simply transferring the aerodynamic load to the inner structural components, has significantly simplified the validation process at subsystem level, in particular with respect to the verification of the elastic properties and the structural stability, durability, and fatigue of the skin over a wide temperature range. Contrarily, dedicated tests would have been necessary to qualify a more innovative and highly anisotropic load-carrying conformal skin, both individually and combined with the inner structure to characterize its actual integrated behavior. Nevertheless, in order to reach the expected L/D improvements and wing box structural mass savings through the maneuvers loads alleviation, the authors believe that the developed finger-like concept has to be complemented by a suitable conformal skin concept to cover the full spectrum of aerodynamic and mechanical requirements faced by morphing structures. However, this still seems difficult to be implemented in practice without a suitable compliant skin technology.

Finally, morphing devices are more prone to aeroelastic instabilities than more conventional architectures integrating passive components. Due to the augmented degrees of freedom and the mutual interaction of the subsystems, especially in malfunctioning or failure conditions, a further maturation of the adaptive winglet technology is centered around an assessment of the aeroelastic properties of the wing assembly, since the preliminary design stages by considering appropriate levels of redundancy in the winglet design. Active shape measurement and control through minimally invasive sensors in both span and chord wise directions may contribute to enabling a smoother shape control with benefits on aircraft load alleviation capabilities and further fuel burn and weight reductions.

### 5.3. Impact on the Adaptive Twist Blade Development

The TRL analysis pointed out the gaps existing between the current level of maturity of the adaptive blade twist concept and its industrialization. Focusing the attention on the different subsystems, it is possible to identify issues and room for maneuver. Starting from the core of the system, that is to say the SMA actuator, the following aspects are faced, strictly related to the nature of the material:The repeatability of the actuation performance: this aspect concerns with the dependence of the material on the cycles of activation. To mitigate this aspect, a specific training operation must be addressed according to a consolidated procedure. It is also important to foresee actuator replacement events within the maintenance plan.The load bearing contribution: a key aspect of the system that strongly impacts its functional and structural performance is the capability of the SMA element to bear loads together with the surrounding structure. This would deliver a lighter, more flexible structure, working in synergy with the SMA element. The current system, due to the tightening safety restrictions of the testing facilities, was conceived to absorb loads without the contribution of the SMA element.The fatigue behavior; the SMA element is integrated into the structure with a certain pre-load to generate enough martensitic phase, fundamental for the authority of the system. This operation has an impact both on the structure and on the SMA itself since it generates a stress field, even without external loads. Even if the current design is compliant with the safety restriction of the testing plants, a more refined design would be necessary to keep the stress level below the fatigue threshold.

Another critical element is the skin. This part, similar for other morphing systems, must transmit the external loads to the interior structure and allow the achievement of the morphed shape through an adequate flexibility. The current solution, suited for demonstration, cannot be applied to an industrial product because of:The stress concentration: even though the current solution allows morphing and the safe execution of the demonstration, the stress concentration at the interface with the inner structure must be adequately addressed, in line with a fatigue-free design, as just mentioned for the structure and the SMA element. Corrugated skins with differential stiffness could be a viable solution to address this problemThe impact proof behavior: civil and military applications of the blade twist system cannot spare design constraint relevant to hail and ballistic events. Adequate skin reinforcement, compliant to the morphing functionality, or an impact proof optimized design of the main interior structure could be considered to address this point.

Still, about the skin element, other critical aspects are the integration with ice protection systems, IPS. In this regard, the heater coils used to activate the SMA could prevent ice accretions by keeping the skin warm, especially close to the leading-edge line where thermal mitigation is more effective than other actions, [48,49]. However, the large flexibility of the skin should be properly considered for those IPSs that imply an intimate connection with the skin itself.

The heating coil is another important subsystem which much can be further explored. The most important aspect here is the heating effectiveness; to improve this parameter and, thus, further increase the power saving performance, the following aspects should be addressed:Interface with the SMA element; any gap should be avoided to maximize the heat conduction. Considering the large flexibility of the system and the large centrifuge actions, the use of thermally conductive pastes is not recommended for the outward projection of material. A valuable solution could be the close coupled integration of the heating coil on the SMA element. This approach, in fact, should not hinder or limit the actuation since the torque rigidity of the coil is negligible with respect to the other structural elementsOutward heat radiation and convection; these types of transfer could be mitigated through a reflecting sheeting deposited on the inner surface of the skin, with thermally insulating properties. These properties, however, should be adequately weighted by the requirement of a warm surface for IPS purposes.

The sensing subsystem must be also revised. The sensing plays a critical role for two main reasons: the structural health monitoring and the functionality of the system. The structural monitoring is particularly critical both for the specific operational role of the system, that is to say a rotating blade under strong centrifuge and aerodynamic actions, and also for the above-mentioned pre-stress level. The functionality of the system is instead strongly influenced by the capability of sensing the actual shape. The current sensing network is slated to a logic able to distinguish between the flap and the twist contribution and, in this sense, meets the requirement of shape reconstruction. However, considering the harsh environment, redundancy is needed to prevent from network unacceptable performance/accuracy drop in terms of structural monitoring and shape reconstruction. Moreover, the structural support used for the sensor and its specific working, that is to say the beam sliding within a guide, must be revised to avoid unwanted friction and jamming of the mechanism.

Last but certainly not the least, the interface with the hosting rotorcraft. Three main types of interfaces can be distinguished: mechanical, electrical (actuator/sensor supply and data acquisition), and data transmission.

Mechanical. The system presents interfaces for the connection of the different bays in serial way, to form a segment of blade of a certain length. The connection of this segment to the original blade implies the design and realization of a new, dedicated interface able to correctly transmit loads.Electrical. The current electrical design must be revised to meet the rotorcraft supply features; this will impact the cabling type and layout and will lead to the introduction of specific electrical units within the circuit (relays, fuses, and so on).Transmission protocol. Because of the narrow available room and the rotation of the blade, the transmission of data coming from the sensors must be accurately faced. Wireless approach seems a viable solution, but this imposes the integration of dedicated elements and the use of dedicated protocols.

For the sake of clearness, the synoptic Table 4 has been prepared to collect the just mentioned aspects and relevant potential approaches. With reference to the maturation path, and considering the abovementioned efforts on the different subsystems, the main tasks can be identified:Redesign of the system following the abovementioned guidelines and prescriptions. A critical design review will end this task with the assessment of possible criticalities; safety, integration and industrialization aspects will represent an important part of this phase and will constitute the basis for the planning of the tests.Manufacturing of a new prototype and procurement of adequate tools for characterization.Ground test demonstration of a new prototype, designed according to the guidelines and the lesson learned from the previous experimental campaigns (laboratory test and WT and WhT demos); the scope of this event is to characterize the enhanced architecture and support the certification phase to obtain the permit to fly.Flight test demonstration of the concept. A final version of the concept will be realized on the basis of the lesson learned by the ground tests. The concept will be integrated on the reference system and flight tests will be held.

After the flight tests, the industrialization aspects will be faced to assure a production in line with the regulations in force. To this aim, all the aspects relevant to engineering, planning, realization, qualification, and monitoring and control of the procedures will be faced and assessed. The engineering task will target the abatement of the production costs, the effectiveness of the processes, the quality of the product, the impact of the processes on the environment, and the customer satisfaction. A backbone scheduling will be established including the main tasks also provided with branches flexible enough to face unexpected events (supplier delays, production line modifications). Effective methods will be selected and will constitute the basis of the planning. Then, the manufacturing will be performed. In this task the technical drawings will be elaborated jointly to the information provided by the engineering on special processes. On this basis, the most appropriate processes will be defined, including the procurement strategy, the production routing, and the inspections. In this way a first item will be manufactured. Inspections and quality analyses will give enough evidence of the compliance of processes and product to the requirements. Finally, after having assessed all criticalities and possible non-conformities, the series production will start. The just mentioned industrialization task will be revised and updated to assure a consistency with the current market requirements and regulations

## 6. Conclusions

The development of a technology typically moves from a scientific basis of knowledge to realize engineering products that can fulfil a set of specifications. Understanding the maturity level of a technology during complex system development is very critical for making good decisions about design, development, and integration, thus influencing the related technology roadmap. The most widely used tool for such maturity assessment is the technology readiness level (TRL) scale. This tool is structured in a number of questions that make it easy to classify the technological product in different ways: analytical modelling, numerical modelling, experiments, interactions with the commissioner, software aspects, production aspects, and so on, allowing to see the investigated system under a complex and complete vision. The engineers and researchers are forced to verify the degree of knowledge associated to the realization more than focusing on the results they attained, which could easily hide the robustness of the logical architecture behind the prototype/demonstrator. This is an exceptional advantage for the researchers and the technologists involved in the realization of novel or even “breakthrough” systems. Indeed, while “evolutionary” realization could take advantage of assessed methods and processes to evaluate the correct set-up of the related architectures, this does not apply, for evident reasons, to innovative concepts that can be just partially covered by previous experiences and expertise.

This study herein reports the assessment of some selected morphing concepts, originally developed by the authors in different research frames. The info used for the evaluation of their maturity derive exclusively by the applicable bibliography, i.e., the publications that the authors performed in the respective programs. Therefore, they do not deal with further details that could have been reached but have been not made available to the general public for reasons of confidentiality toward the participating consortia duly referred. Despite that, this analysis allows attaining important considerations on the development of the architectural ideas since the confidential information deals with the specific application, while the general concepts should have by their nature a universal validity (at least, theoretically).

The study thus highlighted some gaps between the current level of maturity and the industrialization of the developed morphing wing devices. The analysis proved to be extremely interesting because it allowed individuating several critical items that undoubtedly needed to be better addressed for the full maturity of the concepts. In particular, the tool allowed classifying in an ordered manner the strengths, weaknesses, and perspectives of single architectures, giving a complete scenario of their development status. Without such a tool, the analysis would have been more complex, indeed, with a high risk of neglecting some key aspects.

Undoubtedly, speaking of TRL, an un-accurate investigation could easily lead to wrong conclusions. The idea of satisfying simple statements to attain a certain level of maturity cannot be true, and it is the sound outcome of this investigation. There are some aspects where a certain technology can be considered mature, while others are at the preliminary stages. This becomes more accurate as the complexity of the referring system grows. In fact, it can be made of many sub-systems, components, and parts, each with its own readiness level. In turn, this can be further articulated as manufacture, software, and technology (in the strict sense) are considered. Moreover, many other aspects should be further considered or structured, so that the chain could enlarge even more. On this side, it should be added that a growing complexity of the TRL evaluation could not lead to better estimates, so a reasonable compromise should always be searched for (the characteristic time of the TRL evaluation should not overcome the characteristic time of the technology development).

In the proposed technology readiness assessment, both the morphing flap and the adaptive winglet demonstrators have benefited from a more conventional design of rigid body mechanisms with inherent actuators. Overall, the actuation kinematics can thus be considered mature enough to be deployed in actual applications and, most importantly, the comprehensive failure and hazard analyses, with input from industry, showed that no critical safety concerns are associated with the implemented solutions. Moreover, the use of segmented skin concepts rather than suitable compliant skins resulted in more reliable production processes, that however are still considered as MRL 6. Concerning the SMA blade twist architecture, work needs to be done to fix some issues strictly related to the SMA material. Among the others, aspects as the repeatability and the stress concentration for actuation purposes deserve specific attention even if examples of use of SMA for flight applications are already present. Besides the novel materials, the system poses specific problems of integration into preexisting structures. Due to the highly integrated nature of the morphing system, different subsystems are involved. First of all, the mechanical connections to the pre-existing blade resulted in an open issue. On the one hand, the safety, particularly tightening for a rotary structure, imposes robust and effective connections; on the other hand, the economic impact of the installation requires solutions minimally invasive. Then the electrical connections and transmission protocol play their role, being necessary to transmit and handle additional information relevant to the structural safety and the current level of actuation of the morphing system. The not answered questions of the TRL questionnaire dramatically contributed to defining the arrival point of the research to date addressed and to sketch the target of the next industrialization activities. Thus, for each subsystem, maneuver margins and approaches were sketched as basis of discussion for a detailed industrialization plan.

In this work, what appears extremely interesting is that the TRL tool herein used does not find any obstacle in relating with such a relatively new technology such as morphing, along with both its main subdomains: kinematic and compliant arrangements. First, this ensures a sort of “neutrality” of the TRL tool with respect to the technology innovation; this is a relevant result. Second, the fact to have dealt with both the architecture types is essential. If the only kinematic systems had been approached, the trivial consideration to have used the TRL tool with a just evolutionary rather than revolutionary concept would have led to an almost obvious attainment. However, since compliant structures were also investigated, very different in terms of architectures from the kinematic ones, this objection cannot be held.

It is also partially surprising as the status of development of the different architectures is at comparable levels, even with their own peculiarities. Should a general law be searched for, it could be stated that the evolution of a specific technology (morphing in this case) does not proceed by independent steps (whatever is the application), but progresses with continuity, globally. In other words, the time of development of a certain technology seems generally to be a characteristic one (with sporadic spikes, every now and then). This is partially due to the fact that different technologies are in principle available for the subsystems of a morphing concept. For the actuation, in fact, designers may rely upon conventional and innovative technologies; the same for the structure that can be constituted by a kinematic chain or a compliant mechanism, or for the sensing network, based on several type of sensors. The combination of such a variety of technologies and subsystems and in particular their harmonization inevitably leads to a final global TRL, that is a smooth compromise among the single assessment levels. In this sense, the TRL growth remains peculiar of the specific concept, but falls down a general trend. Moreover, the classical development path a morphing system undergoes implicitly leads to a leveling process driven by the specific requirements imposed by the demonstration. This means that the concepts passing the gate, even different from each other, will fall down a certain TRL dispersion, compliant to a gradual growth. In this sense, it may be concluded that the adaptive architectures herein investigated—a morphing (kinematic) flap, a morphing (kinematic) winglet, and a morphing (compliant) helicopter blade—stay commonly around 5, with significant progress in the TRL 6 region. Both the adaptive winglet and the compliant helicopter blade passed the major steps of maturity at TRL6. The analysis did not take care of the manufacturing and SW aspects, since the projects the activities addressed focused on the feasibility of the concepts and on the demonstration of their capabilities in altering performance. Further studies will address the following: to pass the flight demonstration gate, the assessment of materials and processes, and the maturation of the logic of control, to fit the test requirements from one side and to fill the assessment gap with the already developed subsystems on the other side.

## Figures and Tables

**Figure 1 biomimetics-08-00024-f001:**
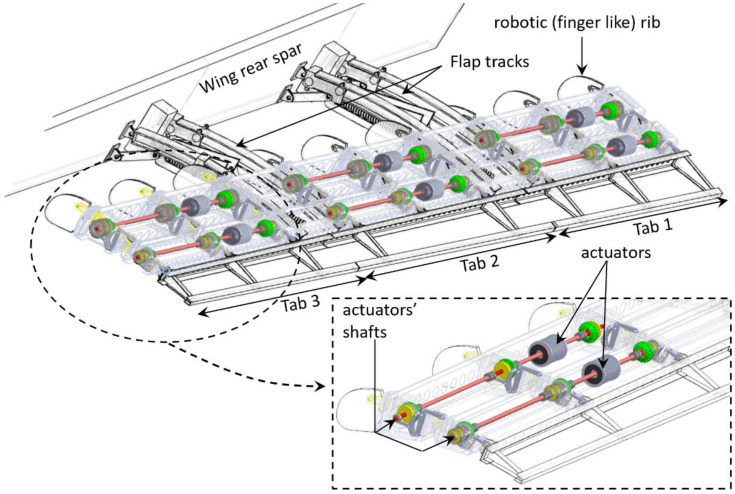
General layout of CleanSky2 Camber Morphing Flap.

**Figure 2 biomimetics-08-00024-f002:**
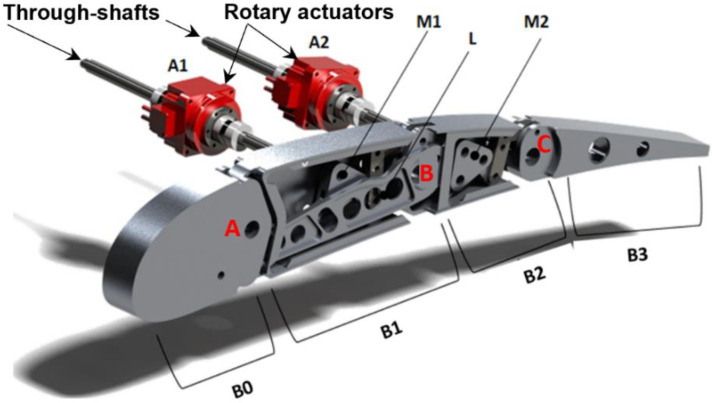
Morphing flap, finger-like robotic rib, and electromechanical actuators.

**Figure 3 biomimetics-08-00024-f003:**
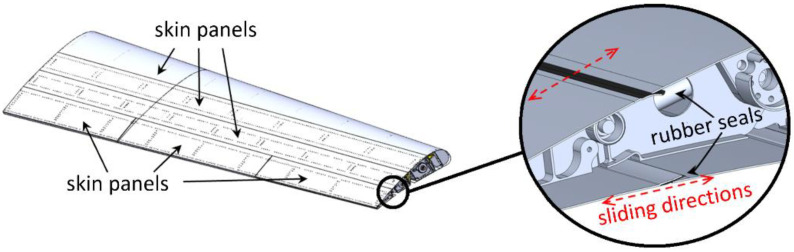
Morphing flap, segmented skin layout.

**Figure 4 biomimetics-08-00024-f004:**
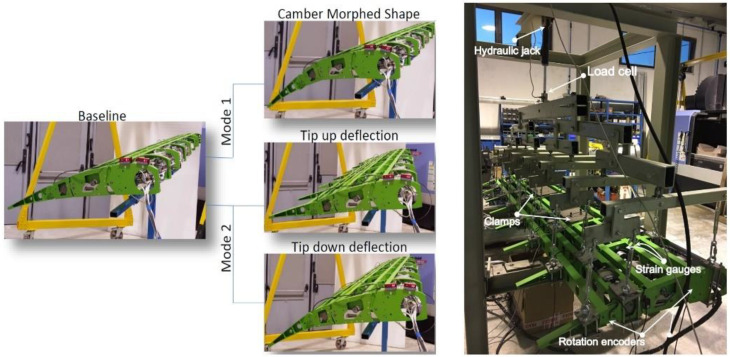
Morphing flap, full-scale functionality test (**left**), and static test (**right**).

**Figure 5 biomimetics-08-00024-f005:**
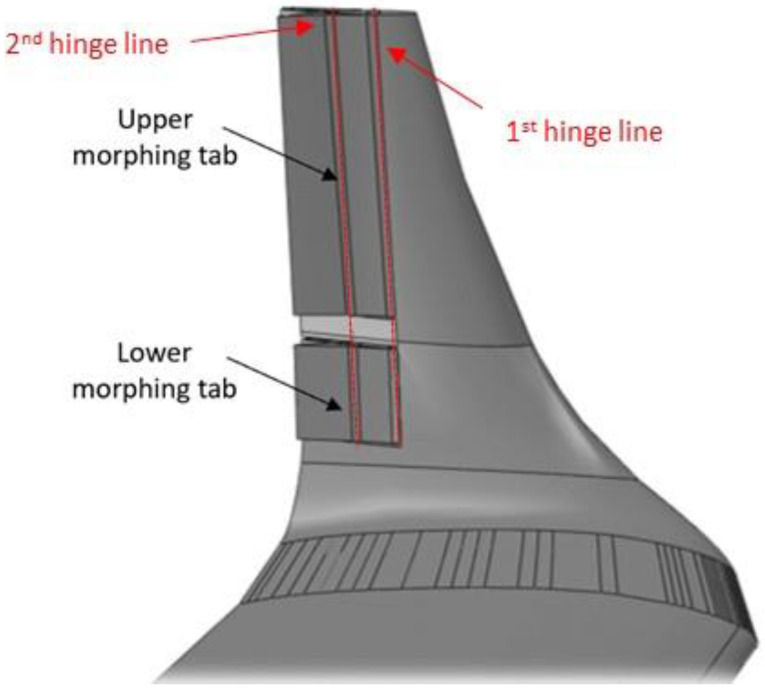
Sketch of the morphing winglet [28]. Figure taken with permission from authors.

**Figure 6 biomimetics-08-00024-f006:**
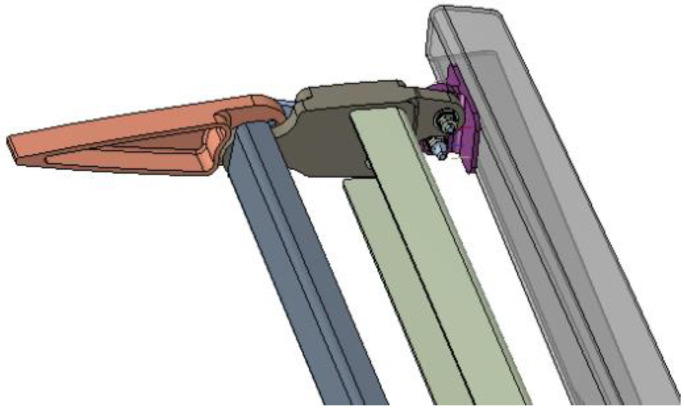
Finger-like mechanism of the upper morphing tab of the winglet, [34]. Figure taken with permission from authors.

**Figure 7 biomimetics-08-00024-f007:**
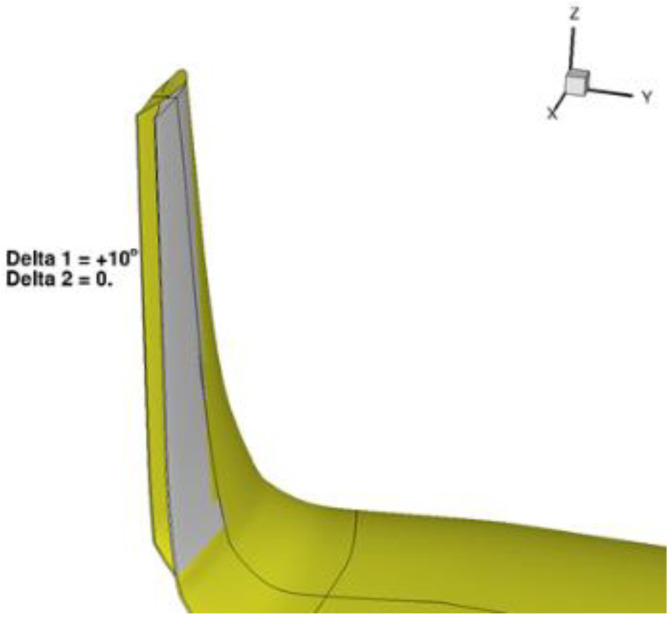
Synchronous deployment of the upper and lower morphing tabs of the winglet, [34,37]. Figure taken with permission from authors.

**Figure 8 biomimetics-08-00024-f008:**
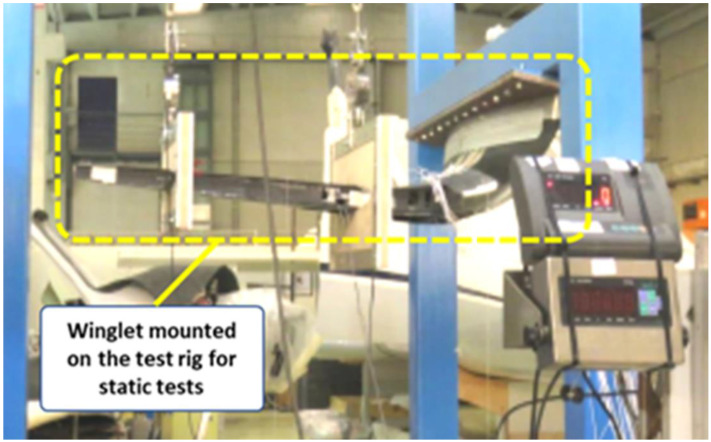
The full-scale demonstrator of the adaptive winglet (mounted on the test rig (Reprinted/ adapted with permission from Ref. [38] Copyright © 2022).

**Figure 9 biomimetics-08-00024-f009:**
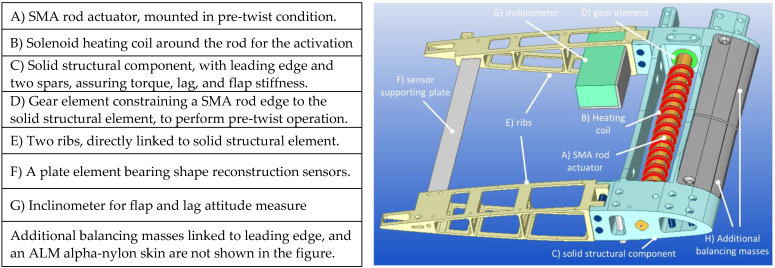
Main components of the adaptive twist blade architecture.

**Figure 10 biomimetics-08-00024-f010:**
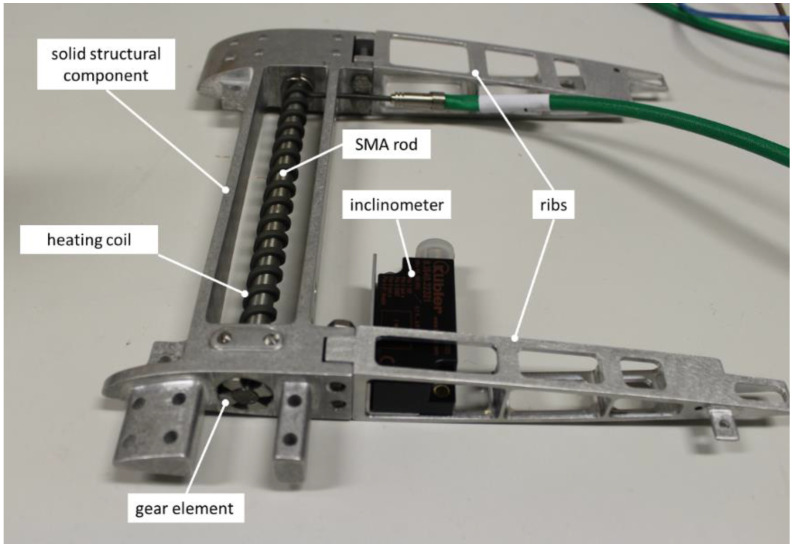
Single frame of the SMA twist architecture.

**Figure 11 biomimetics-08-00024-f011:**
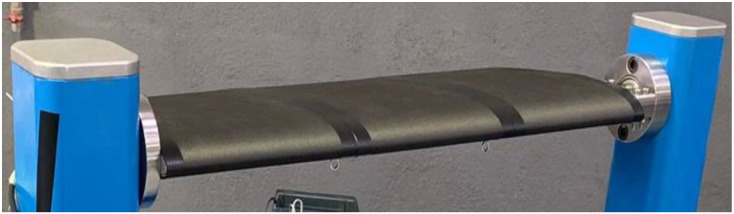
Blade structure constituted by three bays in serial way.

**Figure 12 biomimetics-08-00024-f012:**
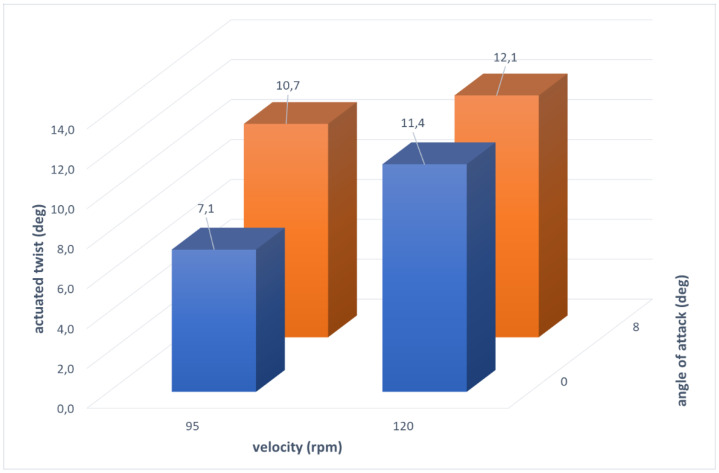
Twist of the blade corresponding to a span of 5 m against velocity and angle of attack.

**Figure 13 biomimetics-08-00024-f013:**
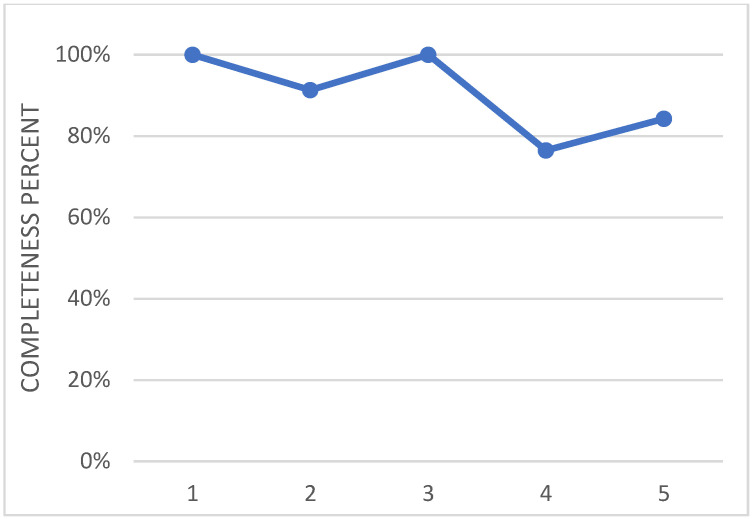
Overall level of maturity of the technology of the adaptive flap actuation system.

**Figure 14 biomimetics-08-00024-f014:**
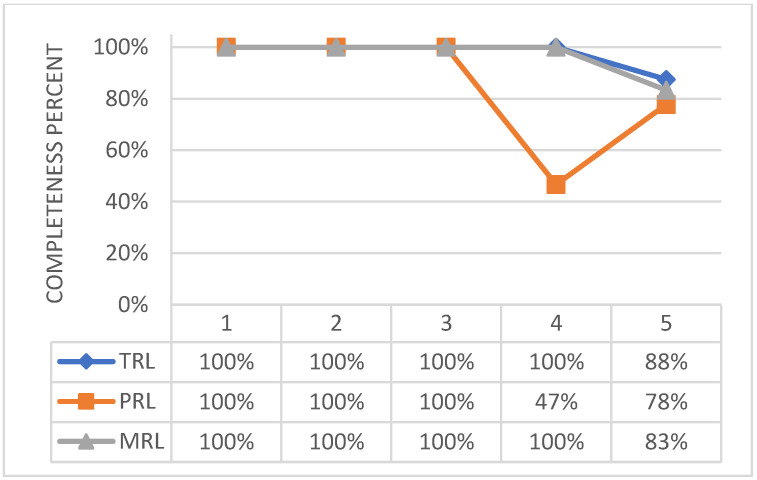
Technological, programmatic, and manufacturing level of maturity (b) of the adaptive flap actuation system.

**Figure 15 biomimetics-08-00024-f015:**
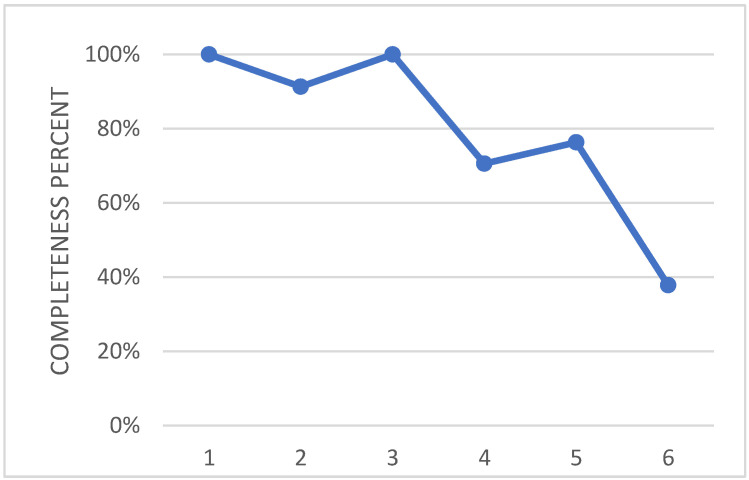
Overall level of maturity of the adaptive winglet actuation system.

**Figure 16 biomimetics-08-00024-f016:**
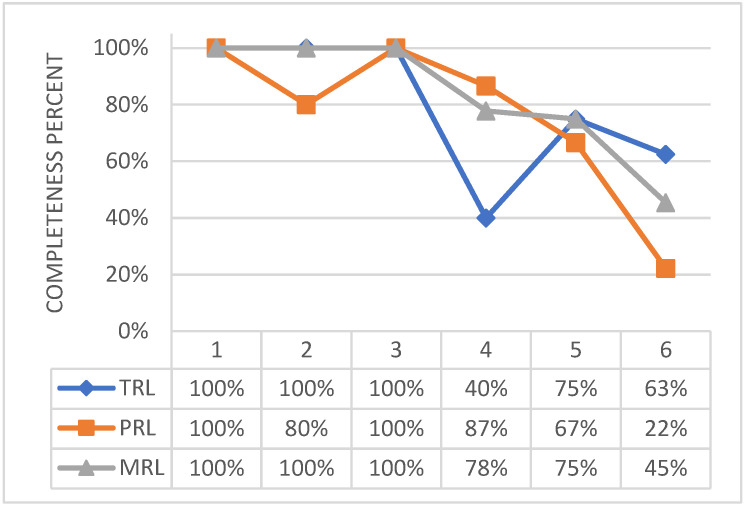
Technological, programmatic, and manufacturing level of maturity of the adaptive winglet actuation system.

**Figure 17 biomimetics-08-00024-f017:**
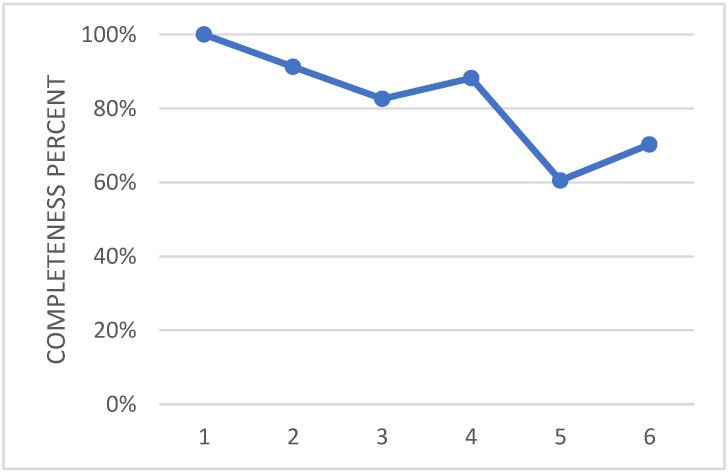
Overall level of maturity of the shape adaptive blade actuation system.

**Figure 18 biomimetics-08-00024-f018:**
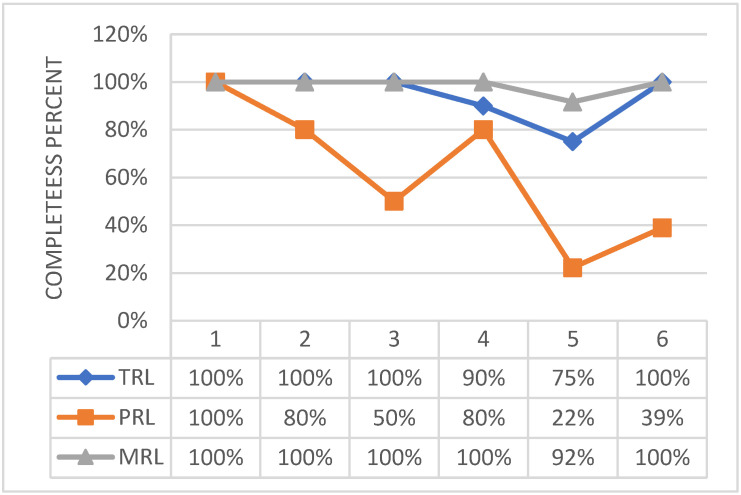
Technological, programmatic, and manufacturing level of maturity of the shape adaptive blade actuation system.

**Table 1 biomimetics-08-00024-t001:** Summary of the queries per technological, programmatic, and maintenance readiness levels, further classified in HW and SW ones.

Transitional Readiness Level		1	2	3	4	5	6	7	8	9
TRL	HW	-	4	4	4	1	1	2	-	-
	SW	4	3	8	9	9	8	1	1	-
	Both	5	9	7	6	7	7	6	8	4
PRL	HW	-	-	1	-	5	2	2	-	-
	SW	-	-	-	1	1	2	2	3	-
	Both	3	10	7	15	13	16	1	2	5
MRL	HW	-	-	4	8	11	9	13	7	5
	SW	-	-	-	1	-	-	1	-	-
	Both	-	-	-	1	1	2	1	1	-
TOTAL		12	26	31	45	48	47	29	22	14

**Table 2 biomimetics-08-00024-t002:** Morphing flap, Impacted development aspects and proposed approaches.

Subsystem	Development Aspect	Proposed Approach
Structure	Manufacturing/assembly time for series production	Design optimization (followed by experimental tests aimed at evaluating the soundness of the proposed solutions)
	Fatigue life	Determination of load spectra (analysis and/or tests), numerical simulations, and test on critical structural items; Implementation of a smart sensing system monitoring the shape and the integrity of the device
	Inspection and continuous maintenance	Digital mock-up analysis, design optimization relying upon advanced software tools for the simulation of relevant inspection/maintenance processes
Actuation, sensing and control	Interface with aircraft equipment	Cable routing design and definition of the electrical links with aircraft equipment and onboard power supply
	Communication protocols with the cockpit and software integration at aircraft level	Flap control software design and test of flap-cockpit communication protocols through advanced flight simulators
Overall device	Demonstration of compliance with Appendix K of EASA CS-25 (interaction of system and structures	Fault and hazard analysis of the system and definition of the necessary redundancies at subcomponents level

**Table 3 biomimetics-08-00024-t003:** Adaptive winglet, impacted development aspects, and proposed approaches.

Subsystem	Development Aspect	Proposed Approach
Structure	Structural design and architecture	Simultaneous shape and topology optimization of the inner skeleton suitable for a conformal morphing skin solution
	Part count reduction	System redesigned in terms of part count, architecture, and materials subject to series production and lifecycle costs constraints thereby improving product performance
	Manufacturing/assembly time for series production	Definition of a suitable manufacturing and assembly procedure
	Inspection and continuous maintenance	Definition of a suitable maintenance plan based on the simulation of relevant inspection/maintenance processes
Morphing skin	highly reliable and technologically mature compliant skin solution that simultaneously deforms and carries loads	Studies on and fatigue resistance, environmental longevity, toughness, abrasion, and chemical resistance Honeycomb structure with carbon rod reinforcement Reinforced corrugated structure with elastomeric surface
Actuation and control	Demonstration of gust alleviation capabilities	Gust alleviation law design and simulations for performance predictions, failure scenarios, comparison of different control approaches
	Feedforward Load control during A/C maneuvers	Control system capability moved from adaptive FF architectures, based on pre-defined positions, to augmented real-time FB control included into the overall aircraft avionics.

**Table 4 biomimetics-08-00024-t004:** Adaptive twist blade, impacted development aspects, and proposed approaches.

Subsystem	Development Aspect	Proposed Approach
SMA actuator	Repeatability of the actuation performance	Specific training operation Actuator replacement events within the maintenance plan
	Load bearing contribution	Optimized structural design to exploit the capability of the SMA element to absorb loads and to keep the stress level below the fatigue threshold
	Fatigue behavior	
Skin	Stress concentration	Corrugated skins with differential stiffness
	Impact proof behavior	Skin reinforcement Impact proof optimized design of the main interior structure
	Integration with IPS	Accurate design of the interface with potential IPS Use of the heater coils to warm the skin
Heating coil	Heating effectiveness (heat leakage)	Close coupled integration of the heating coil on the SMA element to avoid gaps Mitigation of outward heat radiation and convection through reflecting sheeting deposited on the inner surface of the skin, with thermally insulating properties
Sensing	Harsh environment for sensors	Redundancy of the sensors in the layout
	Structural support of sensors	Optimized design to avoid jamming due to the centrifuge-caused friction
Interfaces	Mechanical, load transmission between the blade and the SMA twist	Design and realization of a dedicated interface able to correctly transmit loads
	Electrical, compliant to the rotorcraft supply features	Electrical layout review with insertion of specific electrical units (relays, fuses, adaptors…)
	Sensor data transfer protocol	Wireless communication systems

## Data Availability

Data used in this study are not publicly available for confidentiality reason. Nevertheless, data portions could be available after specific request addressed to the authors.

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
