# Peer review of "A Preliminary Technology Readiness Assessment of Morphing Technology Applied to Case Studies"

_biomimetics, 2023, doi:10.3390/biomimetics8010024_

Round 1
Reviewer 1 Report
This work should be a conference paper or a general presentation. My comments are given below.
ABSTRACT
Needs to be significantly smaller without the initial too general part and any mention of AFRL calculator v2.2.. no one knows what you are talking about. Say technology readiness level instead of TRL, acronyms should not be used in the abstract.
INTRO
Errors (double commas and so on) please avoid such a long romantic introduction. No need to start from the wright brothers. This paper suggests that you are almost the only ones at doing this, I don’t think it is the case. Literature review needs to be more to the point and with much more citations. Line 92 “not so common” why don’t you discuss it?? The last part from 109 to 142 is repeated again in Section 2. It is not clear what the authors want to do.
SHORT DESCRIPTION
Figure 1 is pivotal to the discussion; however it is not that clear, it is hard to understand what is in between the bays. The picture seems to be a snapshot. The discussion is too detailed for an engineering overview and too qualitative for a useful analysis. Figure 4 is not functional to the description.
Line 268 “Such a test campaign proved the ability of the winglet to sustain the limit loads”. Why? What is crucial about this? What is the takeaway? What are pros and cons of this configuration?
Line 290 Why is this information important? I know that it is important, but is it really in this context?
Line 343: Too qualitative
Line 353: “Highly effective” what is a good effectiveness
Figure 13 is confusing. Why the need of normalizing for such a technical manuscript? This should be reduced to a 2D plot, error bars should be added.
SECTION 3
Line 387: “Illuminating statements” sounds a bit too qualitative
Line 397: “There is no surprise”
Section 3.1 can be reduced to few sentences
Section 3.2 Frankly, no one cares about the history of this tool
Line 432: “In the principal menu” why giving this information??
Section 3.3: “Operational hints” it is becoming a user manual!
No technical insight into the structure of the code is given. It sounds like a black box.
SECTION 4
How was the data in Figure 14 evaluated?
All the data presented in this section do not have a scientific base.
Thank you for your time
Sincerely
Author Response
The authors Would like to thank the Reviewer for the time spent reviewing the manuscript.
A point-by-point answers and modifications chart is provided in the attachment.
Looking forward to hearing from You, we send our kindest regards.
The authors.

Reviewer 2 Report
1. There are numerous errors in citations. These have been indicated in the attached pdf. Lots of typos as well should be corrected.
2. Narrative is quite verbose and could be made far more concise. Recommend tightening up the writing.
3. The conclusions are not conclusions. Please write a set of conclusions. Is this a good tool? Does it represent the readiness levels appropriately? Are these morphing technologies ready for implementation, and if not why not? How did the tool help you assess this?

Author Response
The Authors would like to thank the Reviewer for the time spent reviewing the manuscript.
A point-by-point answers and modifications chart and provided in the attachment.
Looking forward to hearing from You, we send our kindest regards.
The Authors

Reviewer 3 Report
Overall, the paper is interesting, and should be of interest to other researchers. The paper does not describe any novel engineering developments that have not been previously published, but it reviews some leading technologies and makes use of an existing tool (the TRL calculator) in a novel way. In that regard, this is not a highly novel, or ground-breaking paper, but I think it will be interesting. This paper might be more appropriately published in a journal dedicated to systems engineering or program management.
The paper version I reviewed had various sections of text highlighted in different colors. I assume these are changes that were made since the earlier draft?? In any case, the entirety of the introduction is so rife with grammatical errors as to be incomprehensible on the first read-through. I had to read these paragraphs multiple times in order to decipher the intended meaning. This is in start contrast to other sections of the paper which are well-written. I suspect that different authors penned different sections, and/or the highlighted sections are new text that was not as thoroughly reviewed. I have uploaded images of various pages in which I found numerous errors. I can't imagine trying to write a paper in Italian, so I have great sympathy and admiration for the authors who must publish this paper in English. That said, it cannot pass in its current state.
A few technical matters:
1. You present a set of figures that state the assessed readiness levels of the 3 technologies. I don't understand why the TRL shown in Figs 13, 15, and 17 don't match that shown in Figs 14, 16, and 18, respectively. This should be explained, or the figures corrected. For the odd-numbered figures, the caption says "Overall" maturity, so is it meant to be a combination of TRL, PRL and MRL? If so, please remove the "TRL" that is printed on these figures.
2. In the conclusion, especially the last paragraph, you make several claims that are not substantiated by evidence. Specifically, I object to the claim that "the evolution of a specific technology (morphing in this case), does not proceed by independent steps (whatever is the application), but progresses with continuity, globally." You have not provided sufficient evidence for this. The conclusion is not the appropriate place for hypotheses.
2a. Further, this statement is completely inappropriate for an archival paper: "The analysis did not take care of manufacturing and SW aspects, but it is believed that a further study would have again confirmed the commonality of the developments and attained a level globally around 4." This is purely opinion with no basis in fact, and should not be part of a scientific report.
Overall, I found the paper interesting, and I look forward to reviewing a corrected version.
Author Response
The Authors would like to thank the Reviewer for the time spent reviewing the manuscript.
A point-by-point answers and modifications chart is provided in the attachment together with a copy of the manuscript with the requested revision.
Looking forward to hearing from You, we send our kindest regards.
The Authors
